# Interpersonal responses to facial expressions of disgust, anger, and happiness in individuals with varying levels of social anxiety

**Marije aan het Rot** ◉*, **Christina Friederici, Sandra C. Krause**◉**, Peter J. de Jong**◉

Department of Psychology, University of Groningen, Groningen, Netherlands

* m.aan.het.rot@rug.nl

**Data Availability Statement:** The anonymized data are freely accessible via DataverseNL using the following DOI: https://doi.org/10.34894/LZKWBR.

## Abstract

### Background and objectives

Facial expression recognition has been studied extensively, including in relation to social anxiety. Nonetheless, a limited number of studies examined recognition of disgust expressions. Results suggest that disgust is perceived as more threatening than anger, and thus may invite more extreme responses. However, few studies have examined responses to facial expressions. These studies have focused on approach-avoidance responses. Our primary aim was to examine to what extent anger and disgust expressions might invite interpersonal responses in terms of quarrelsomeness-agreeableness and dominance-submissiveness. As social anxiety has been previously associated with a heightened sensitivity to anger and disgust expressions, as well as with alterations in quarrelsomeness-agreeableness and dominance-submissiveness, our secondary aim was to examine whether social anxiety would moderate these responses.

### Methods

Participants were 55 women and 43 men who completed social anxiety measures, including the Brief Fear of Negative Evaluation scale, and two tasks that involved "targets" expressing anger, disgust, or happiness at 0%, 50%, or 100%. Participants first indicated how quarrelsome or agreeable and how dominant or submissive they would be towards each target, and then how much they would avoid or approach each target.

### Results

While 100% disgust and anger expressions invited similar levels of quarrelsomeness and avoidance, 50% disgust invited more quarrelsomeness and stronger avoidance than 50% anger. While these patterns were not meaningfully moderated by social anxiety, individuals with higher BFNE scores showed a relatively strong approach of happy faces.

### Limitations

Actual interpersonal behaviour in response to facial expressions was not assessed.

**Funding:** The authors received no specific funding for this work.

**Competing interests:** The authors have declared that no competing interests exist.

## Conclusions

Findings support the relevance of disgust as an interpersonal signal and suggest that, especially at mild intensity, disgust may have a stronger impact than anger on people's quarrelsomeness and avoidance responses. Findings provided no support for the view that people with social anxiety would be particularly responsive to disgust (or anger) expressions.

## Introduction

Emotion processing has often been examined using emotion recognition tasks. Most of these tasks include emotional facial expressions as stimuli, such as happy and angry faces. However, for our everyday social interactions, not only an accurate recognition of facial expressions is important, but also how we respond to people displaying these expressions. Therefore, studies of how we behave in response to specific facial expressions may complement studies on the recognition of these expressions, and provide vital insights into how facial expressions may guide social interaction. Previous studies have found that happy faces invite approach and agreeable-dominant behaviour whereas angry faces invite avoidance and quarrelsome behaviour [1, 2]. Less is known about responses to facial expressions of disgust. Yet these expressions, similar to expressions of anger or happiness, may affect people's interpersonal responses and might thus have important social implications.

### Disgust versus anger

While there is general consensus that disgust originally evolved as a disease-avoidance mechanism [3], current theories propose that disgust has an important function not only in safeguarding one's physical integrity by avoiding potential contaminants, but also in maintaining the integrity of one's social group by communicating and thereby avoiding socio-moral transgressions by others [4]. When we are asked to transgress their own internalized socio-moral rules, we typically experience feelings of disgust [5]. Such instances of moral disgust are thought to help us keep on the right track and stay connected to our social group [4]. From this perspective, expressions of disgust can be conceptualised as reflecting a morally-motivated social condemnation of someone transgressing a socio-moral rule and a wish for distancing ourselves from the transgressor [6]. As such, disgust expressions represent a signal of social rejection, indicating a request to increase interpersonal distance [7].

While anger expressions are also considered to communicate condemnation, research has demonstrated that these expressions are more likely to be perceived as selfishly motivated than morally motivated [8]. We express anger when another person's behaviour is considered a direct threat to our current goals, whereas we express disgust when the other person's behaviour reflects a violation of a more general social norm [6]. In other words, anger seems to communicate the condemnation of a particular behaviour in a specific situation [9], while disgust seems to communicate the more general condemnation of the transgressor as a person, i.e. in terms of their character [10]. On the one hand, the shared component of condemnation as signalled by anger as well as disgust may invite similar interpersonal responses to both emotion expressions. On the other hand, however, due to the more general character of the condemnation conveyed by disgust expressions, the responses invited by these expressions might be more pronounced than those invited by expressions of anger.

Communal and agentic responses to facial expressions. Most past facial emotion response studies have assessed response in terms of approach-avoidance. Participants in

these studies are usually shown images of various expressions (e.g., angry, happy) and asked to choose to either (a) push a joystick or move a manikin away from them (indicating implicit avoidance), or (b) pull the joystick or move the manikin toward them (indicating implicit approach) [11, 12]. To date, two such studies have included disgust expressions; both studies were conducted in Psychology students. One study found that while expressions of happiness and sadness implicitly invited approach and expressions of anger invited avoidance, expressions of disgust tended not to invite any significant pattern of response [2]. Nonetheless, when participants were explicitly asked whether they would approach or avoid targets displaying specific expressions, they indicated that they would avoid targets expressing disgust. This explicit response was also found in the other study [13], though only when participants were left uncertain about why the emotion was expressed by the target. When the disgust expressions were visually accompanied by a drink as the potential source of the target's expressed emotion, disgust invited approach. This implies that disgust expressions may primarily invite avoidance when perceivers believe themselves to be the source of the emotion.

While the approach-avoidance model provides information about one well-recognized dimension of behaviour [14], a deeper understanding of the sources of variation in responses to emotional expressions may be obtained by considering more than one dimension. Timothy Leary [15] proposed a two-dimensional model for the conceptualization and analysis of interpersonal characteristics. This interpersonal circumplex model continues to be popular [16]. It considers two orthogonal dimensions: communion and agency. In terms of responses to emotion expressions [1], the communion dimension represents the warmth with which individuals might respond, ranging from quarrelsomeness (low communion) to agreeableness (high communion), while the agency dimension represents the power with which individuals might respond, ranging from submissiveness (low agency) to dominance (high agency). While the link between the approach-avoidance model and the interpersonal circumplex model remains to be tested empirically, variation in approach-avoidance is likely explained by variation in both communion and agency.

Studies that employ the interpersonal circumplex model to examine responses to facial expressions complement the aforementioned approach-avoidance studies because these studies can increase insight into why we might approach or avoid others expressing certain emotions. Two past studies used an emotion response task derived from the interpersonal circumplex model [1, 17]. These studies found that happy expressions generally invite agreeable-dominant responses and anger expressions primarily invite quarrelsomeness. While neither study examined responses to facial expressions of disgust, these might also invite quarrelsomeness: Within the interpersonal circumplex, trait inference ratings for disgust and anger expressions have previously been reported to be comparable [18]. Nonetheless, as mentioned before, the responses invited by expressions of disgust might be more pronounced than those invited by expressions of anger. As a consequence, disgust expressions might invite more quarrelsomeness than angry expressions.

## Facial expressions and social anxiety

Individuals diagnosed with social anxiety disorder (SAD) report symptoms that disrupt daily functioning and tend to center around a fear of situations in which they would be exposed to possible scrutiny by others, i.e. a fear of negative evaluation [19]. These individuals worry about others' perceptions and experience high social interaction anxiety [20]. This anxiety seems to elicit the submissive behavior that characterizes individuals with SAD [21–23]. The theoretical understanding of SAD has been focused on the appraisal of threats to the social self

[24]. In line with this, it has been argued that the maintenance of SAD is partially due to the misreading of others' facial expressions [25].

It has also been argued that, to individuals diagnosed with SAD, the most threatening expression is anger [26]. That said, disgust has been studied less often than anger [27] and disgust expressions are more likely than anger expressions to result from a negative evaluation of others' character [10]. As SAD tends to center around a fear of negative evaluation, individuals with this diagnosis may find disgust expressions particularly threatening [28]. Consequently, disgust expressions might influence the behavior of individuals with SAD even more than anger expressions, or might influence their behavior even when disgust is expressed only mildly.

Defensive responses to interpersonal threats are also common in typically developing individuals (5, 21). Thus, insight into the causes of SAD can also be gained from analogue research on how unselected individuals with varying social anxiety levels respond to others' facial expressions. Two past approach-avoidance studies have compared responses to facial expressions in high versus low socially anxious individuals. One study found more implicit avoidance of angry expressions in the high socially anxious individuals, even though explicit avoidance ratings of the anger expressions were similar across the two subgroups [29]. Another study, using "crowd" stimuli containing variable percentages of neutral and emotional facial expressions, found that when the stimuli contained more angry than neutral faces, the high socially anxious individuals showed more implicit avoidance [30]. Similar to the first study, explicit avoidance ratings of the crowds did not significantly differ between the two subgroups. These findings suggest higher levels of social anxiety might be associated with more implicit avoidance of anger expressions. While disgust expressions were not studied, these might also be avoided more. In line with this idea, high socially anxious individuals have been found to rate interactions with targets showing disgust expressions as costlier [31] and, more generally, show heightened sensitivity to disgust experiences [32, 33]. Also, social anxiety has been associated with a negative bias in the processing of emotional facial expressions, including disgust expressions [34].

## The present study

Studies in unselected participants have examined responses to different expressions of emotion either along the dimension of approach-avoidance [2, 13] or along the two dimensions of communion and agency [1, 17]. Responses of socially anxious individuals have only been studied in terms of approach-avoidance [29, 30], and studies to date did not include disgust expressions, which might be particularly salient for socially anxious individuals who are typically high on fear of negative evaluation.

In light of the proposed unique relevance of disgust as a social signal, the primary aim of the present study was to replicate and extend previous work on the interpersonal responses to anger and happy facial expressions [1, 17] by including expressions of disgust. As a second aim, we examined if the responses to facial expressions of happiness, anger, and disgust would be moderated by social anxiety. Social anxiety has been associated with a negative bias in the processing of emotional facial expressions [34]. Nonetheless, how this cognitive component of social anxiety might affect the behaviour of socially anxious individuals in response to others' facial emotional expressions during interaction is still unclear. Examining responses to disgust expressions seems particularly relevant, as socially anxious individuals tend to fear negative evaluation and interpersonal rejection [35], which is communicated not only by anger but also, and perhaps even more so, by disgust [36].

Therefore, in the present study we not only included angry and happy faces as stimuli in a facial emotion response task but also examined communal (quarrelsome-agreeable) and

agentic (dominant-submissive) responses to disgusted faces. Of note, we used another stimulus set than the one previously used to examine responses to angry and happy expressions [1, 17] as a means to test the robustness of our earlier findings. Further, we examined responses not only in terms of communion and agency but also in terms of (explicit) approach-avoidance. This was similar to past approach-avoidance studies [2, 13, 29, 30] and allowed us to explore the link between the approach-avoidance model and the interpersonal circumplex model. Furthermore, we examined how responses varied as a function of trait social anxiety. To this end, we administered the Social Interaction Anxiety Scale (SIAS), which primarily measures the behavioral and emotional features of social anxiety [37], and the Brief Fear of Negative Evaluation (BFNE) scale, which primarily assesses its more cognitive component [38].

Our specific hypotheses centered around responses to disgust versus anger expressions. We tested these hypotheses in an analogue sample; while the magnitude of the responses of unselected participants with varying social anxiety levels may differ from that of individuals diagnosed with SAD, the direction of the responses was expected to be similar [39]. Our first hypothesis was that disgust expressions and anger expressions would both invite quarrelsomeness, yet disgust expressions would invite more quarrelsomeness than anger expressions. This hypothesis was derived from the idea that while anger, which was previously found to invite quarrelsomeness [1, 17], communicates a negative evaluation of another person's behavior in a specific situation, disgust communicates a negative evaluation of the other person's character, which is more general [10].

Our second hypothesis was that higher levels of social anxiety would be associated with more submissiveness in response to both disgust and anger. This is in line with previous research showing that individuals diagnosed with SAD report having problems with high submissiveness [23] and behave more submissively during social interactions than normally developing comparison individuals, particularly when they feel anxious [21] and perceive their interaction partners as quarrelsome [22].

## Materials and methods

### Participants

Participants were 98 first-year Psychology students at the University of Groningen (56% female) who were given partial course credit for their time. Their mean age was 20 years (*SD* = 2). Their ethnicity was not assessed but most University of Groningen students are of Western European descent. The departmental ethics committee reviewed the study protocol (Ethics Committee of Psychology, University of Groningen, protocol number: 13031-NE).

### Measures

Responses to facial expressions were assessed using a computer task that included grayscale face-forward pictures of target faces either with a neutral expression or expressing anger, disgust, or happiness in one of two intensities. Happy faces were primarily included to allow for replication of previous findings with another stimulus set [1, 17]. The neutral and 100% emotion intensity stimuli were identical to the stimuli used in a previous approach-avoidance study [40]. The 50% intensity expressions were generated from the 100% intensity expressions using FotoMorph by Digital Photo Software. Six female and six male faces of individuals of Western European descent were presented for each expression, i.e. 84 faces in total. Each face was presented in a different random order for every participant (500 ms per face), followed by a fixation cross (300 ms) and a response screen (5000 ms).

The language of the task could be adjusted to the participants' mother tongue. There were 3 parts. Each part used the same stimuli but different instructions. In part 1, conforming to

previous studies [1, 17], the response screen included a grid for participants to indicate how they would behave during an interaction with the target. The grid was divided into four quadrants by orthogonal axes. Scores on both axes ranged from -100 to +100 and represented the Communion dimension (from quarrelsomeness on the left to agreeableness on the right) and the Agency dimension (from dominance on the top to submissiveness on the bottom) of the interpersonal circumplex. The middle of the grid represented neutral behaviour. Person-mean scores on Communion and Agency were not significantly correlated, $r$(98) = 0.06, $p > 0.5$.

Part 2 data were not usable due to experiment error, but we describe the task for replication purposes. Prior to the task, participants were instructed to respond to the stimuli by indicating how they thought *the target* would behave *towards them*. In other words, we asked participants to rate the behaviour of the persons displaying the varying facial expressions. Erroneously, however, the question below the response grid in part 1 of the computer task ("How would you behave towards this person?") did not change on-screen during part 2. Thus we did not analyse the part 2 data.

In part 3, participants were asked how much they would respond to each face with avoidance or approach. They could indicate their response by clicking more towards the left on a horizontal line (minimum score: -100) or more towards the right (maximum score: +100). Person-mean approach-avoidance scores were positively correlated with person-mean scores on Communion, $r$(98) = 0.36, $p < 0.0004$, and Agency, $r$(98) = 0.20, $p < 0.05$, indicating that participants with higher approach tendencies are more likely to be communal and agentic towards others.

There were no incorrect responses and no instructions were given on how quickly to respond. The task recorded response times (calculated from the time the response screen was shown) to be able to control for between- and within-person variation in the speed of responding, which might otherwise influence responses. For example, we expected that some participants would take shorter to respond to some expressions, making these responses less likely to have been influenced by explicit cognitive processes.

To assess the cognitive component of social anxiety at the trait level, we administered the 12-item Brief Fear of Negative Evaluation (BFNE) scale [38, 41]. A sample BFNE item is: "I often worry that I will say or do the wrong things." Participants used a 5-point scale ranging from 0 (not at all) to 4 (extremely) to indicate how characteristic each item was of them. Total scores can range from 0 to 48 with higher scores representing more social anxiety.

To assess the more emotional and behavioural components of trait social anxiety, we also administered the 20-item Social Interaction Anxiety Scale (SIAS) [37]. Sample SIAS items are: "I am tense mixing in a group" and "I have difficulties talking with other people." Again, participants used a 5-point scale ranging from 0 (not at all) to 4 (extremely) to indicate how characteristic each item was of them. Total scores can range from 0 to 80 with scores of 34–42 representing subclinical social anxiety and scores of 43 or higher representing clinical social anxiety.

We also administered the Revised Disgust Propensity and Sensitivity Scale [DPSS-R; 42], developed to assess sensitivity to disgust experienced by the self. Its 16 items (e.g., "I avoid disgusting things", "I become disgusted more easily than other people") are equally divided over two subscales. Participants used a 5-point scale ranging from 1 (never) to 5 (always) to indicate how often each item applied to them. Total scores on the Propensity and Sensitivity subscales can both range from 1 to 40 with higher scores representing a higher disgust propensity and a higher disgust sensitivity, respectively.

## Procedures

Participants were recruited using a website only accessible to first-year Psychology students at the University of Groningen. Students interested in the study signed up online for a laboratory visit, during which they received written and verbal information about the study. All participants provided written informed consent. The study was carried out in accordance with the Declaration of Helsinki.

Each laboratory visit took around 45 minutes. Participants first completed the SIAS, BFNE, and DPSS-R (and no other questionnaires) and then completed the computer task. In each part of the task, the experimental condition was the facial expression, i.e. happiness, anger, disgust, or none (neutral). Participants received an oral explanation of part 1 before starting this part, and a written explanation plus a brief computerized training session before starting parts 1–3. The average time to respond to the 84 stimuli was 3 minutes in part 1 and 2 minutes in parts 2 and 3.

## Statistical analyses

The data had a multilevel structure, with repeated measures of responses to various facial expressions of various intensities (within-person variation) in participants with varying social anxiety levels (between-person variation). Therefore, using PROC MIXED in SAS 9.4 (SAS Institute, Cary, NC), we conducted multilevel analyses to test our hypotheses. Multilevel models included a random intercept and the default error covariance matrix. The degrees of freedom for $F$ tests were determined by dividing the residual degrees of freedom into between- and within-person portions. For testing the significance of $F$ and $t$ test outcomes, we used an alpha level of 0.05 unless indicated otherwise below.

To test the hypotheses, analyses were conducted on the data from part 1 of the computer task. When the outcome variable was Communion, positive and negative values indicated agreeableness and quarrelsomeness, respectively. When the outcome variable was Agency, positive and negative values indicated dominance and submissiveness, respectively. We included Response Time (log-transformed due to skewness) as a covariate because response times were previously found to vary by facial expression. Moreover, we included Target Face Identity as a covariate because responses to faces are influenced by more than their expressions, for example by target gender or attractiveness (both captured by Target Face Identity). To be consistent with previous studies [1, 17], we report the outcomes of analyses that included both covariates. Nonetheless, the results did not change when we omitted them.

Models 1 and 2 were used to test the first hypothesis, that communal responses to disgusted faces would be more extreme than communal responses to angry faces. These models were also used to explore whether agentic responses to disgusted faces would be comparable to agentic responses to angry faces. Model 1 only included Expression (4 levels: neutral, angry, disgusted, happy) as a predictor. Model 2 included Expression (3 levels: angry, disgusted, happy), Intensity (2 levels: 50%, 100%), and their interaction as predictors. Model 2 did not include neutral expressions because there were no 50% intensity neutral expressions. Follow-up testing of significant interaction terms was done using $t$ tests and included a Tukey-Kramer correction.

Models 3 and 4 were used to test the second hypothesis, that social anxiety would moderate agentic responses to angry and disgusted faces. These models were also used to explore whether social anxiety would moderate communal responses to facial expressions. BFNE and SIAS scores were examined separately as potential moderators because they are thought to assess different components of social anxiety [37, 38] and indeed were only moderately correlated, see Measures. Model 3 included Expression (3 levels), BFNE (continuous; standardized),

and the Expression by BFNE interaction as predictors. Model 4 included Expression (3 levels: angry, disgusted, happy), SIAS (continuous; standardized), and the Expression by SIAS interaction as predictors. Like Model 2, Models 3–4 did not include neutral expressions because there were no 50% intensity neutral expressions. Besides, Models 3 and 4 did not include Intensity because this would have required testing three-way interaction effects, and our second hypothesis only concerned the two-way interaction between social anxiety and facial expression. Significant interaction effects were examined by testing the difference between pairs of estimated simple slopes, using $t$ tests. Simple slopes estimated for all 3 expressions at high (mean + 1 SD) and low (mean − 1 SD) levels of social anxiety. When interpreting the outcomes of the $t$ tests, a Bonferroni correction was applied such that alpha / 9 = 0.0056.

For exploring approach-avoidance responses to the facial expressions, Models 1–4 were rerun on the data from part 3 of the computer task, with approach-avoidance as the outcome variable. Positive and negative values indicated approach and avoidance, respectively. Additional analyses are presented below.

Please note that reported numbers of degrees of freedom for $F$ and $t$ tests described in the results range widely because these numbers depend on whether a between-person effect, a within-person effect, or an interaction between a between-person effect and a within-person effect is being described. Effect sizes are expressed using Cohen's $d$.

## Results

Table 1 provides descriptive information about the sample. The internal consistency of the BFNE was excellent (Cronbach coefficient alpha of 0.95). The internal consistency of the SIAS was excellent (Cronbach coefficient alpha of 0.91). Both the Propensity subscale and the Sensitivity subscale had acceptable internal consistency (with Cronbach coefficient alphas of 0.78).

### Communal and agentic responses to facial expressions

For each outcome variable, the estimated least-squares means and corresponding standard errors per expression (Model 1) or per expression-intensity combination (Model 2) are provided in Table 2. The $F$ values of the predictors and covariates in Models 1–2 are presented in Table 3. Hypothesis 1 focused on Communion; Agency was an additional outcome variable.

**Communion.** In Model 1, the effect for Expression was significant, $p < 0.001$, $d = 6.3$. Mean levels for communion were positive for happy ($M = 49.50$, $SE = 1.56$) and neutral faces ($M = 13.29$, $SE = 1.74$) indicating agreeableness. The agreeable response towards happy faces was higher than towards neutral faces, $t(291) = 27.28$, $p < 0.0001$, $d = 3.2$. Mean levels for communion were negative for angry ($M = -35.63$, $SE = 1.56$) and disgusted faces ($M = -37.87$,

**Table 1. Descriptive information about the sample (N = 98).**

| | |
|---|---|
| Male: female ratio | 43: 55 |
| Age in years | $M = 20$ ($SD = 2$) |
| BFNE total score | $M = 17$ ($SD = 10$) |
| SIAS total score | $M = 19$ ($SD = 11$) |
| DPSS-R Propensity total subscale score | $M = 22$ ($SD = 4$) |
| DPSS-R Sensitivity total subscale score | $M = 17$ ($SD = 4$) |

BFNE = Brief Fear of Negative Evaluation scale. SIAS = Social Interaction Anxiety Scale. DPSS-R = Revised Disgust Propensity and Sensitivity Scale.

**Table 2. Estimated least-squares means (standard errors) resulting from Models 1 and 2 per outcome Variable.**

| | Communion[a] | Agency[b] | Approach-Avoidance[c] |
|---|---|---|---|
| Model 1: Expression | | | |
| • Neutral | 13.29 (1.74)*** | 9.97 (2.03)*** | -1.88 (1.27) |
| • Angry | -35.63 (1.56)*** | 3.20 (1.89) | -41.87 (1.12)*** |
| • Disgusted | -37.87 (1.56)*** | 6.88 (1.89)*** | -45.04 (1.12)*** |
| • Happy | 49.50 (1.56) *** | 24.21 (1.89)*** | 35.98 (1.12)*** |
| Model 2: Expression*Intensity | | | |
| • Angry 50% | -27.70 (1.81)*** | 3.98 (2.09) | -33.46 (1.29)**** |
| • Angry 100% | -43.59 (1.81)*** | 2.48 (2.08) | -50.22 (1.29)*** |
| • Disgusted 50% | -34.46 (1.81)*** | 7.43 (2.08)*** | -39.31 (1.29)*** |
| • Disgusted 100% | -41.34 (1.81)*** | 6.38 (2.08)** | -50.67 (1.29)*** |
| • Happy 50% | 40.84 (1.80)*** | 21.12 (2.08)*** | 25.32 (1.29)*** |
| • Happy 100% | 58.19 (1.81)*** | 27.33 (2.08)*** | 46.72 (1.29)*** |

*$p < 0.05$ $p < 0.01$

***$p < 0.001$ (Significantly different from zero.) Behaviour was reported in terms of communion and agency (Task 1) or approach-avoidance (Task 3).

[a]Negative and positive values for communion indicate quarrelsomeness and agreeableness, respectively.

[b]Negative and positive values for agency indicate submissiveness and dominance, respectively.

[c]Negative and positive values for approach-avoidance indicate avoidance and approach, respectively.

**Table 3. Facial expression and intensity as predictors of responses.**

| | Communion | Agency | Approach-Avoidance |
|---|---|---|---|
| Model 1 | | | |
| -Covariates | | | |
| • Response Time | 1.44 | 6.86** | 14.89*** |
| • Target Face Identity | 23.47*** | 1.98* | 57.80*** |
| -Predictor | | | |
| • Expression[a] | 2933.09*** | 150.04*** | 4085.67*** |
| Model 2 | | | |
| -Covariates | | | |
| • Response Time | 2.09 | 6.69** | 12.10*** |
| • Target Face Identity | 16.72*** | 1.89* | 49.31*** |
| -Predictors | | | |
| • Expression[b] | 4542.99*** | 215.64*** | 6415.19*** |
| • Intensity | 4.53* | 1.95 | 11.48** |
| • Expression$^2$ by Intensity | 136.16*** | 8.09*** | 324.45*** |

Values represent outcomes of *F*-tests.

*$p < 0.05$,

**$p < 0.01$,

***$p < 0.001$.

[a]Four levels: angry, disgusted, happy, neutral.

[b]Three levels: angry, disgusted, happy. Behaviour was reported in terms of communion (quarrelsomeness-agreeableness) and agency (submissiveness-dominance) or approach-avoidance.

$SE = 1.56$), indicating quarrelsomeness. Quarrelsome responses to these two expressions did not differ significantly, $t(291) = -2.08$, $p > 0.17$, $d = -0.24$.

Nonetheless, in Model 2, the effects for Expression, $p < 0.001$, $d = 9.7$, Intensity, $p < 0.05$, $d = 0.43$, and the Expression by Intensity interaction, $p < 0.001$ were all significant. Communion increased from 50% happy expressions, $M = 40.84$, $SE = 1.80$, to 100% happy expressions, $M = 58.19$, $SE = 1.81$, $t(194) = 11.78$, $p < 0.0001$, $d = 1.7$, indicating more agreeableness with more intense happy expressions. Communion decreased from 50% angry expressions, $M = -27.70$, $SE = 1.81$, to 100% angry expressions, $M = -43.59$, $SE = 1.81$, $t(194) = -10.78$, $p < 0.0001$, $d = -1.5$, and from 50% disgusted expressions, $M = -34.46$, $SE = 1.81$, to 100% disgusted expressions, $M = -41.34$, $SE = 1.81$, $t(194) = -4.67$, $p < 0.0001$, $d = -0.67$, indicating more quarrelsomeness with more intense anger and disgust expressions. While quarrelsomeness towards 100% angry versus 100% disgusted expressions did not differ significantly, $t(194) = -1.53$, $p > 0.64$, $d = -0.22$, quarrelsomeness was higher towards 50% disgusted expressions than towards 50% angry expressions, $t(194) = 4.58$, $p < 0.0002$, $d = 0.66$.

Overall, disgust and anger expressions both invited quarrelsomeness, yet mild (50% intensity) disgust expressions invited more quarrelsomeness than mild anger expressions. This supported hypothesis 1.

**Agency.** In Model 1, the effect for Expression was significant, $p < 0.001$, $d = 1.4$. Mean levels for agency were not significantly different from zero for angry faces, $M = 3.20$, $SE = 1.89$. Mean levels for agency were positive for happy ($M = 24.21$, $SE = 1.89$), neutral ($M = 9.97$, $SE = 2.03$), and disgusted faces ($M = 6.88$, $SE = 1.89$), indicating dominance. Thus, while disgust invited dominance, anger did not, $t(291) = 3.48$, $p < 0.004$, $d = 0.41$. All four means were significantly different from each other, except for the difference between dominant responses to neutral versus disgusted expressions, $t(291) = 2.39$, $p > 0.08$, $d = 0.28$.

In Model 2, while the effect for Intensity was not significant, $p > 0.05$, $d = 0.28$, the effect for Expression, $p < 0.001$, $d = 2.1$, and for the interaction were, $p < 0.001$ (see Table 2). Agency increased from 50% happy expressions, $M = 21.12$, $SE = 2.08$, to 100% happy expressions, $M = 27.33$, $SE = 2.08$, $t(194) = 4.09$, $p < 0.001$, $d = 0.59$, indicating more dominance with more intense happy expressions. Agency did not significantly change from 50% angry expressions, $M = 3.98$, $SE = 2.09$, to 100% angry expressions, $M = 2.48$, $SE = 2.08$, $t(194) = -0.98$, $p > 0.92$, $d = -0.14$, and from 50% disgusted expressions, $M = 7.43$, $SE = 2.08$, to 100% disgusted expressions, $M = 6.38$, $SE = 2.08$, $t(194) = -0.69$, $p > 0.98$, $d = -0.10$, indicating unchanged dominance with more intense angry or disgusted expressions. The difference between dominant response to angry versus disgusted expressions did not reach significance either at 50%, $t(194) = 2.27$, $p > 0.21$, $d = 0.33$, or at 100%, $t(194) = 2.56$, $p > 0.11$, $d = 0.37$.

The outcomes of Model 2 analyses are summarized in Fig 1. Responses to increasingly happy expressions were increasingly agreeable-dominant. Responses to increasingly angry expressions were increasingly quarrelsome, without becoming more or less dominant. Similarly, responses to increasingly disgusted expressions were increasingly quarrelsome without becoming more or less dominant. Nonetheless, in terms of quarrelsomeness, participants appeared more sensitive to mild disgust expressions than to mild anger expressions.

## Role of social anxiety

BFNE and SIAS scores were only moderately positively correlated, $r(98) = 0.56$, $p < 0.0001$, which fits with the idea that the questionnaires assess different components of social anxiety [37, 38]. The mean (SD) total BFNE and SIAS scores are provided in Table 1. Their range was 0–46 and 2–51, respectively. According to the SIAS, clinical levels of social anxiety were present in 6 participants (5 female) and subclinical levels in 3 participants (1 female).

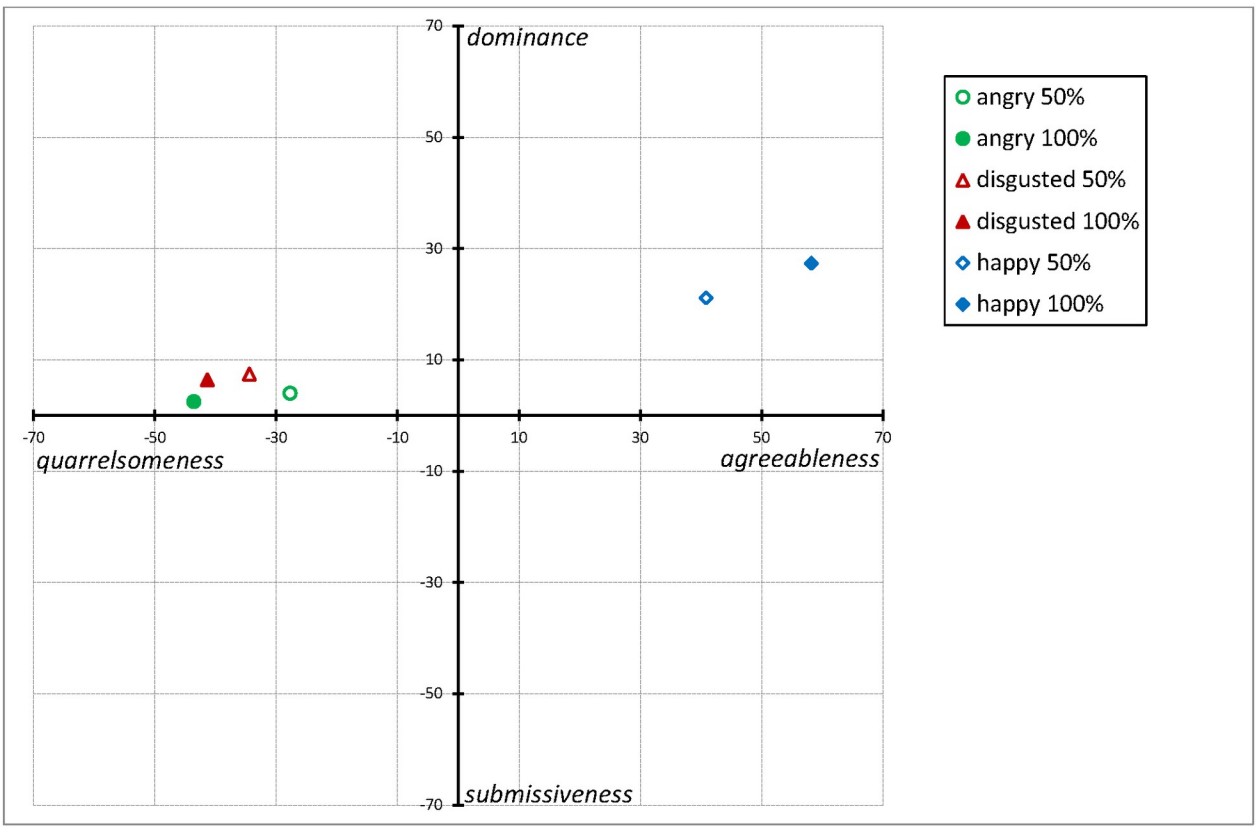

**Fig 1. Responses to 50% and 100% angry, disgusted, and happy facial expressions in terms of quarrelsomeness-agreeableness and submissiveness-dominance.** Quarrelsomeness-agreeableness represents communion (horizontal dimension) and submissiveness-dominance represents agency (vertical dimension). Horizontal and vertical error bars have been omitted for clarity.

The *F* values of the predictors and covariates in Models 3–4 are presented in Table 4. Hypothesis 2 focused on Agency; Communion was an additional outcome variable.

**Communion.** The Expression by BFNE interaction was significant, $p < 0.001$. However, in post-hoc tests, higher BFNE scores did not significantly predict higher communion in response to disgusted faces, i.e. less quarrelsomeness, $b = 2.89$, $t(6872) = 1.75$, $p > 0.08$, $d = 0.04$. Also, BFNE scores did not significantly predict communion in response to angry faces, $b = 2.69$, $t(6872) = 1.63$, $p > 0.10$, $d = 0.04$, nor in response to happy faces, $b = -0.73$, $t(6872) = -0.44$, $p > 0.65$, $d = 0.01$.

The Expression by SIAS interaction was significant, p < 0.001. However, follow-up testing revealed similar results to those for BFNE in the previous paragraph. In post-hoc tests, higher SIAS scores did not significantly predict higher communion in response to disgusted faces, i.e. less quarrelsomeness, $b = 2.74$, $t(6872) = 1.65$, $p > 0.09$, $d = 0.04$. Also, SIAS scores did not significantly predict communion in response to angry faces, $b = 2.17$, $t(6872) = 1.31$, $p > 0.19$, $d = 0.03$, nor in response to happy faces, $b = -1.81$, $t(6872) = -1.09$, $p > 0.27$, $d = 0.03$.

**Agency.** The Expression by BFNE interaction was significant, $p < 0.01$, but the simple slopes for disgust, $b = -2.34$, $t(6872) = -1.20$, $p > 0.23$, d = 0.03, anger, $b = -2.67$, $t(6872) = -1.37$, $p > 0.16$, d = 0.03, and happiness, $b = 0.76$, $t(6872) = 0.39$, $p > 0.69$, d = 0.01, were not. The Expression by SIAS interaction was not significant, $p > 0.05$. Apparently, agentic responses were not moderated by social anxiety.

**Table 4. Facial expression and social anxiety as predictors of responses.**

|  | Communion | Agency | Approach-Avoidance |
|---|---|---|---|
| Model 3 |  |  |  |
| -Covariates |  |  |  |
| • Response Time | 1.37 | 8.15** | 15.38*** |
| • Target Face Identity | 16.11*** | 1.89* | 45.28*** |
| -Predictors |  |  |  |
| • Expression[a] | 4373.81*** | 214.76*** | 1351.18*** |
| • BFNE | 1.11 | 0.59 | 0.34 |
| • Expression by BFNE | 7.29*** | 6.13** | 23.77*** |
| Model 4 |  |  |  |
| -Covariates |  |  |  |
| • Response Time | 1.41 | 6.89** | 15.37*** |
| • Target Face Identity | 16.11*** | 1.89* | 45.01*** |
| -Predictors |  |  |  |
| • Expression[a] | 4378.31*** | 214.99*** | 1443.10*** |
| • SIAS | 0.45 | 2.06 | 0.01 |
| • Expression by SIAS | 10.81*** | 1.74 | 1.40 |

Note. Values represent outcomes of *F*-tests.

*$p < 0.05$,

**$p < 0.01$,

***$p < 0.001$.

[a]Three levels: angry, disgusted, happy. BFNE = Brief Fear of Negative Evaluation scale scores. SIAS = Social Interaction Anxiety Scale scores. Behaviour was reported in terms of communion (quarrelsomeness-agreeableness) and agency (submissiveness-dominance) or approach-avoidance.

Overall, social anxiety did not significantly moderate communal and agentic responses to happy, angry, or disgusted expressions. Thus, we found no support for hypothesis 2, that social anxiety would be negatively associated with agentic responses to disgust and anger.

### Approach-avoidance responses

See Tables 2–4 again. In Model 1, the effect for Expression was significant, $p < 0.001$, $d = 7.5$. Neutral faces were neither approached nor avoided, $M = -1.88$, $SE = 1.27$. Happy faces were approached, $M = 35.98$, $SE = 1.12$. Angry ($M = -41.87$, $SE = 1.12$) and disgusted faces ($M = -45.04$, $SE = 1.12$) were avoided. All means differed from each other. Most relevant to our study, disgusted faces were avoided more than angry faces, $t(291) = 3.77$, $p < 0.002$, $d = 0.44$.

In Model 2, the effects for Expression, $p < 0.001$, $d = 11.5$, Intensity, $p < 0.01$, $d = 0.69$, and their interaction, $p < 0.001$, were all significant. Happy expressions were approached more at 100% intensity, $M = 46.72$, $SE = 1.29$, than at 50% intensity, $M = 25.32$, $SE = 1.29$, $t(194) = 18.67$, $p < 0.0001$, $d = 2.7$. More intense angry expressions were avoided more, 100% versus 50%: $M = -50.22$, $SE = 1.29$, versus $M = -33.46$, $SE = 1.29$, $t(194) = 14.63$, $p < 0.0001$, $d = 2.1$. More intense disgust expressions were also avoided more, 100% versus 50%: $M = -50.67$, $SE = 1.29$, versus $M = -39.31$, $SE = 1.29$, $t(194) = 9.91$, $p < 0.0001$, $d = 1.4$. However, while 100% disgusted and angry expressions were avoided to a similar degree, $t(194) = 0.39$, $p > 0.99$, $d = 0.06$, 50% disgusted expressions were avoided more than 50% angry expressions, $t(194) = 5.10$, $p < 0.0001$, $d = 0.73$, see Fig 2.

In Model 3, the Expression by BFNE interaction was significant, $p < 0.001$. Higher BFNE scores predicted more approach of happy expressions, $b = 3.91$, $t(6936) = 3.34$, $p < 0.0009$,

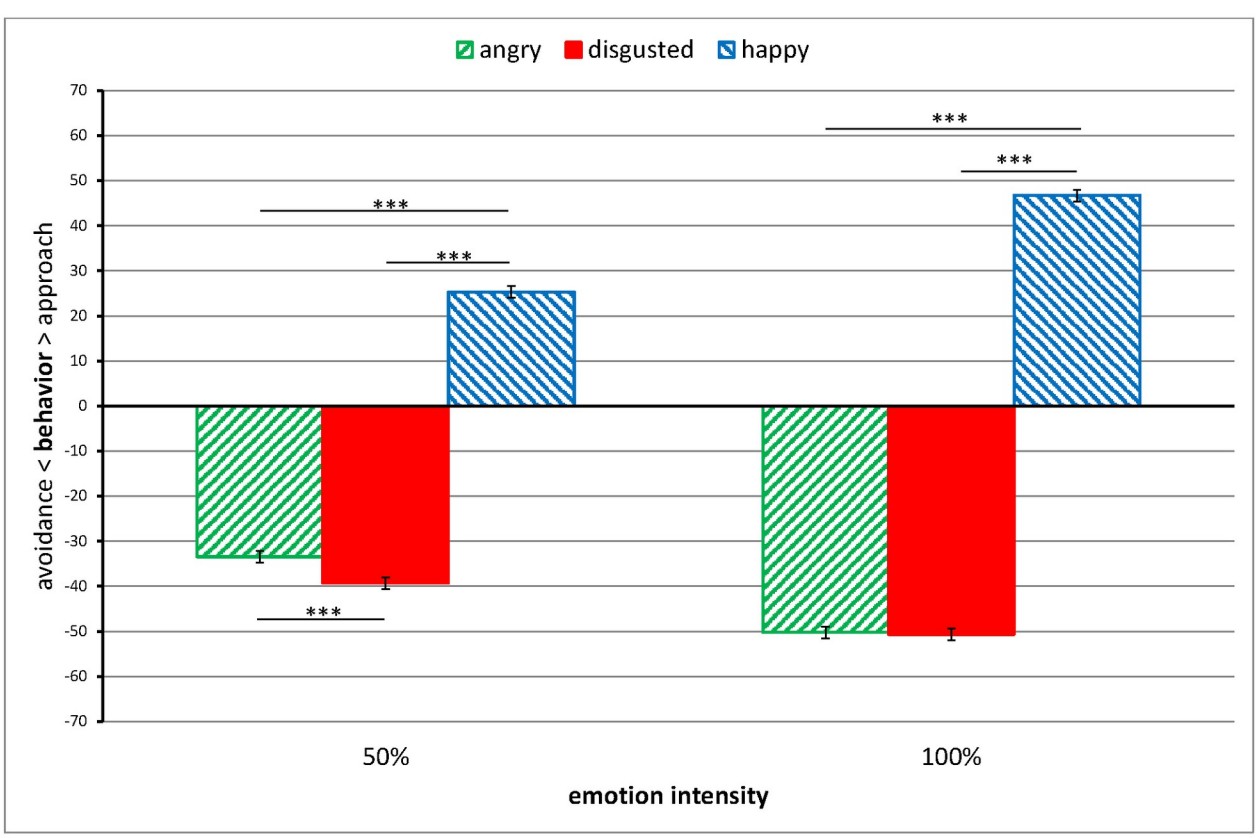

**Fig 2. Responses to 50% and 100% angry, disgusted, and happy facial expressions in terms of approach-avoidance.** Error bars represent standard errors. ***p < 0.001.

$d$ = 0.08. BFNE scores were not significantly associated with approach-avoidance of anger expressions, $b$ = -0.38, $t(6936)$ = -0.32, $p$ > 0.74, $d$ = 0.01, or disgust expressions, $b$ = -1.68, $t(6872)$ = -1.43, $p$ > 0.15, $d$ = 0.03. In Model 4, the Expression by SIAS interaction was not significant, $p$ > 0.05.

In sum, mild disgust expressions were avoided more than mild anger expressions. Higher levels of social anxiety were not significantly associated with more avoidance of either disgust or anger.

## Discussion

We studied communal, agentic, and approach/avoidance responses to facial expressions of disgust, anger, and happiness, and examined if these responses were moderated by social anxiety. The major findings can be summarized as follows: (1) Whereas happy faces invited agreeableness (i.e., high communion) and approach, both disgust and anger expressions invited quarrelsomeness (i.e., low communion) and avoidance. However, mild (50% intensity) disgust expressions invited more quarrelsomeness and avoidance than mild anger expressions. (2) In terms of agency, while happy faces invited more dominance than disgust and anger expressions, there was no significant difference in invited dominance between disgust and anger expressions (for both 50% and 100% expression intensity stimuli). (3) While social anxiety did not significantly moderate agentic or avoidant responses to either disgust or anger expressions, happy faces invited more approach of happy faces in individuals with higher BFNE scores.

While the findings that happiness invited agreeable dominance and that anger primarily invited quarrelsomeness replicate previous studies [1, 17], the present study focused on responses to disgust expressions. In relation to our first hypothesis, we found that disgust expressions invited quarrelsomeness and mild dominance, as did anger expressions. We also found that while there was no significant difference in terms of dominance, responses to mild disgust expressions were more quarrelsome (but not more dominant) than responses to mild anger expressions (Fig 1). Thus, in terms of communion, we may behave differently when others express disgust than when others express anger. We might be more sensitive to mild disgust expressions than to mild anger expressions. This appears in line with the idea that disgust expressions are thought to communicate a general aversion of someone's character, while anger expressions more likely result from a situational interpersonal threat [10, 27].

Both disgust and anger expressions also invited avoidance (Fig 2). Similarly, disgust and anger expressions invited avoidance in a previous study [2]. However, we only found this avoidance to be comparable between the two facial emotions for intense expressions; for mild expressions, disgust was avoided more than anger. This is in line with the results for quarrelsomeness as discussed in the previous paragraph. As suggested in the Introduction, while the shared component of condemnation as signalled by both anger and disgust may invite similar interpersonal responses to both emotion expressions, due to the more general character of the condemnation conveyed by expressions of disgust [6, 10], the responses invited by these expressions might be more pronounced than those invited by expressions of anger. Additionally, it has been argued that disgust is more likely to invite avoidance when participants think they are the source of the expression [13]. This may have been relevant in the present study as participants were specifically instructed to indicate how they would respond someone with a disgusted (or other) facial expression during an interaction with that person.

As already mentioned in the first paragraph of this Discussion, both anger and disgust expressions invited mild dominance. However, in terms of dominance-submissiveness (i.e., along the dimension of agency), we did not find that more socially anxious individuals were more likely to respond with submissiveness to others expressing disgust. This was not in line with our second hypothesis. Similarly, we did not find that more socially anxious participants were more (or less) likely to show avoidance of disgust expressions. While socially anxious individuals have been found more likely to avoid angry faces [29, 30], past studies did not include disgust stimuli. Ultimately, the question of how trait social anxiety might moderate responses to anger and disgust expressions deserves further study. For example, it is unclear if our findings would generalize to individuals with SAD, which is characterized by high levels of submissive behavior [21–23]. Notably, these individuals are mostly likely to behave submissively during social interactions in which state anxiety is high [21]. This means that trait social anxiety might moderate responses to anger and disgust expressions mostly at times when individuals feel anxious. Unfortunately, we did not assess state anxiety in the present study.

Nonetheless, there was one additional finding that involved trait social anxiety. Specifically, individuals with higher BFNE scores were more likely to report approach of happy expressions. While it is unclear why the BFNE moderated this behavioural response whereas the SIAS did not, it may be relevant that the BFNE assesses the more cognitive component of social anxiety [38, 41] and was only moderately correlated with the SIAS. In any case, the present finding appears somewhat in line with prior research, also cited in the previous paragraph, in which individuals with SAD were more likely than controls to report agreeableness during interactions in which they felt emotionally secure [21].

We addressed some limitations of previous studies using the same computer task [1, 17]: we only included participants whose mother tongue matched the language of the task and we used face stimuli that were more recent than the previously used Picture of Facial Affect

Series [43]. One remaining task-related limitation is that we assessed how individuals might behave in response to facial expressions presented as static images of unknown targets, rather than how individuals actually behave towards real-life others, who are often no strangers. However, this is also done in approach-avoidance studies [2, 13, 29, 30]. Another task-related limitation is that we assessed responses to emotional expressions without verifying whether these expressions were recognized accurately. However, we also did this in our past studies and we note that while anger tends to be misinterpreted as disgust and vice versa, there is no evidence that a lower resolution of the expression affects disgust recognition more than anger recognition [44]. Besides, emotion recognition does not require the conscious processing of facial expressions [45] and is not required for appropriate social interaction.

There were some additional limitations. Firstly, we did not test whether all 50% intensity expressions, generated using a digital morphing technique, were perceived as such by participants. However, a systematic variance in the perceived intensity level of 50% versus 50% disgust expressions is at odds with the finding that these two types of expressions did not systematically invite different levels of agency. Secondly, there may have been carryover effects across the task, during which participants were repeatedly exposed to the same emotional faces. Increased facial familiarity has been associated with both decreased and increased neural responses [46, 47] as well as with altered facial preferences [48]. However, while this may have particularly affected our results on approach-avoidance (which was examined after communion and agency), increased familiarity cannot explain our finding of differential responses to anger versus disgust expressions.

As a final note, we did not assess implicit responses to facial expressions, but explicitly asked participants for their responses. In previous research [2], while anger expressions invited avoidance (and happy expressions invited approach) both explicitly and implicitly, disgust expressions invited avoidance at an explicit level but not at an implicit level. Thus, our results on explicit responses to disgust expressions may not translate to implicit responses to disgust expressions.

## Conclusions

We replicated past findings on behavioural responses to happiness and anger [1, 17]. Moreover, we found that mild disgust expressions may invite more quarrelsomeness and avoidance than mild anger expressions. Besides, while trait social anxiety did not significantly moderate behavioural responses to anger or disgust, it may be positively associated with approach in response to happy expressions.

## Author Contributions

**Conceptualization:** Marije aan het Rot, Peter J. de Jong.

**Data curation:** Christina Friederici, Sandra C. Krause.

**Formal analysis:** Marije aan het Rot.

**Supervision:** Marije aan het Rot.

**Writing – original draft:** Marije aan het Rot.

**Writing – review & editing:** Marije aan het Rot, Christina Friederici, Sandra C. Krause, Peter J. de Jong.

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
