## [Decision Letter · Decision Letter 0]

21 Dec 2021

PONE-D-21-32367Interpersonal responses to facial expressions of disgust, anger, and happiness in individuals with varying levels of social anxietyPLOS ONE

Dear Dr. aan het Rot: Thank you for submitting your manuscript to PLOS ONE. After careful consideration, we feel that it has merit but does not fully meet PLOS ONE’s publication criteria as it currently stands. Therefore, we invite you to submit a revised version of the manuscript that addresses the points raised during the review process.

 Please follow suggestions of REviewer 1 in your revision, and address all issues in your rebuttal letter  to me.

We look forward to receiving your revised manuscript.

Kind regards,

Marina A. Pavlova, PhD

Academic Editor

PLOS ONE

Journal Requirements:

2. Please note that according to our submission guidelines (http://journals.plos.org/plosone/s/submission-guidelines), outmoded terms and potentially stigmatizing labels should be changed to more current, acceptable terminology. To this effect,  “Caucasian” should be changed to “white” or “of [Western] European descent” (as appropriate)

Reviewers' comments:

Reviewer's Responses to Questions

**Comments to the Author**

1. Is the manuscript technically sound, and do the data support the conclusions?

Reviewer #1: Yes

2. Has the statistical analysis been performed appropriately and rigorously? 

Reviewer #1: Yes

3. Have the authors made all data underlying the findings in their manuscript fully available?

Reviewer #1: Yes

4. Is the manuscript presented in an intelligible fashion and written in standard English?

Reviewer #1: Yes

5. Review Comments to the Author

Reviewer #1: The manuscript is well written, covering an interesting topic, and the study is technically sound. The authors investigated communal, agentic, and approach-avoidance responses to different intensity levels (i.e., 0%, 50%, 100%) of facial expressions of emotions, with the purpose to test the hypotheses that disgust and anger emotions would elicit quarrelsomeness response in the subjects, disgust more than anger. The moderation role of social anxiety has been also investigated. Participants completed social anxiety measures and two tasks of social cognition in which they were invited to indicate how quarrelsome-agreeable, dominant-submissive they would be towards faces expressing emotions, and how much they would avoid-approach the stimuli. The authors demonstrated that disgust and anger expressions at maximum intensity (i.e., 100%) produced similar levels of quarrelsomeness and avoidance, whereas disgust at 50% of intensity produced more quarrelsomeness and greater avoidance than anger at 50%. Social anxiety did not moderate agentic or avoidant responses to disgust and anger expressions, but participants with higher social anxiety showed more approach to happy faces.

To make the manuscript suitable to be accepted I list some minor concerns to be addressed:

- In the Results section, the test significance (i.e., p) should be reported also in the text, together with descriptive data, such as mean and standard deviation (e.g., in the first sentences of the paragraphs Communion and Agency of the Results section).

- In the Measures section, more information should be reported, specifically: the total score, the total number of items, and the direction of the scale (i.e., what a greater score means) of the Brief Fear of Negative Evaluation (BFNE) scale; the total number of items of the Social Interaction Anxiety Scale (SIAS); the total score, the direction of the scale, and the exact number of items for each subscale (it is not clear if the total items are equally divided between the two subscales) of the Revised Disgust Propensity and Sensitivity Scale (DPSS-R).

- Also, in the Measures section, internal consistency and correlational results should be moved in the Results section.

- The Revised Disgust Propensity and Sensitivity Scale (DPSS-R) seems not to be included in any of the analyses. However, the DPSS-R mean and standard deviation could be relevant to be reported. The authors could consider adding an additional table showing demographic characteristics of the sample as well as DPSS-R scores.

6. PLOS authors have the option to publish the peer review history of their article (what does this mean?). If published, this will include your full peer review and any attached files.

Reviewer #1: No

---

## [Author Response · Author response to Decision Letter 0]

10 Jan 2022

See Response to Reviewers file (attached)

---

## [Decision Letter · Decision Letter 1]

2 Feb 2022

Interpersonal responses to facial expressions of disgust, anger, and happiness in individuals with varying levels of social anxiety

PONE-D-21-32367R1

Dear Dr. aan het Rot:

We’re pleased to inform you that your manuscript has been judged scientifically suitable for publication and will be formally accepted for publication once it meets all outstanding technical requirements.

Kind regards,

Marina A. Pavlova, PhD

Academic Editor

PLOS ONE

Additional Editor Comments (optional):

Reviewers' comments:

Reviewer's Responses to Questions

**Comments to the Author**

1. If the authors have adequately addressed your comments raised in a previous round of review and you feel that this manuscript is now acceptable for publication, you may indicate that here to bypass the “Comments to the Author” section, enter your conflict of interest statement in the “Confidential to Editor” section, and submit your "Accept" recommendation.

Reviewer #1: All comments have been addressed

2. Is the manuscript technically sound, and do the data support the conclusions?

Reviewer #1: Yes

3. Has the statistical analysis been performed appropriately and rigorously? 

Reviewer #1: Yes

4. Have the authors made all data underlying the findings in their manuscript fully available?

Reviewer #1: Yes

5. Is the manuscript presented in an intelligible fashion and written in standard English?

Reviewer #1: Yes

6. Review Comments to the Author

Reviewer #1: I believe that the manuscript is ameliorated after the authors addressed all my concerns, and that it is suitable to be accepted.

7. PLOS authors have the option to publish the peer review history of their article (what does this mean?). If published, this will include your full peer review and any attached files.

Reviewer #1: No